# Prevalence of Pathogenic *Leptospira* spp. in Non-Volant Small Mammals of Hutan Lipur Sekayu, Terengganu, Malaysia

**DOI:** 10.3390/pathogens11111300

**Published:** 2022-11-05

**Authors:** Nur Juliani Shafie, Najma Syahmin Abdul Halim, Adedayo Michael Awoniyi, Mohamed Nor Zalipah, Shukor Md-Nor, Mohd Ulul Ilmie Ahmad Nazri, Federico Costa

**Affiliations:** 1Faculty of Science and Marine Environment, Universiti Malaysia Terengganu, Kuala Nerus 21030, Terengganu, Malaysia; 2Institute of Biology, Federal University of Bahia, Salvador 40170-115, Brazil; 3Institute of Collective Health, Federal University of Bahia, Salvador 40110-040, Brazil; 4Department of Biological Sciences and Biotechnology, Faculty of Science and Technology, Universiti Kebangsaan Malaysia, Bangi 43600, Selangor, Malaysia; 5STEM Foundation Centre, Universiti Malaysia Terengganu, Kuala Nerus 21030, Terengganu, Malaysia; 6Gonçalo Moniz Research Center, Oswaldo Cruz Foundation, Brazilian Ministry of Health, Salvador 41745-715, Brazil; 7Department of Epidemiology of Microbial Diseases, Yale School of Public Health, New Haven, CT 06511, USA; 8Lancaster Medical School, Lancaster University, Lancaster LA1 4YW, UK

**Keywords:** animal reservoirs, leptospirosis, recreational area, rodents, Malaysia

## Abstract

Leptospirosis is an important zoonotic disease that is transmitted worldwide through infected small mammals such as rodents. In Malaysia, there is a paucity of information on the animal reservoirs that are responsible for leptospirosis transmission, with only a few studies focusing on leptospirosis risk in recreational areas. Therefore, in this study we characterized the species composition and the prevalence of pathogenic *Leptospira* spp. in non-volant small mammals of Hutan Lipur Sekayu, Terengganu. We performed ten trapping sessions totaling 3000 trappings between September 2019 and October 2020. Kidney samples from captured individuals were extracted for the PCR detection of pathogenic *Leptospira* spp. Overall, we captured 45 individuals from 8 species (1.56% successful trapping effort), with 9 individuals testing positive for pathogenic *Leptospira*, that is, a 20% (n = 9/45) prevalence rate. *Rattus tiomanicus* (n = 22) was the most dominant captured species and had the highest positive individual with pathogenic *Leptospira* (44.4%, n = 4/9). Despite the low successful trapping effort in this study, the results show the high diversity of non-volant small mammals in Hutan Lipur Sekayu, and that they could also maintain and transmit pathogenic *Leptospira*.

## 1. Introduction

The Southeast Asian rainforests are the earth’s oldest, with countless biodiversity [1], and the majority of rainforests in this region are considered biodiversity hotspots [2]. Malaysia has one of the world’s twelve most biologically diverse countries [3] and supports extensive diversity of fauna. Its mammalian diversity is noteworthy, with approximately 440 species, of which about 15% are endemic [4]. Additionally, there are approximately 185 species of non-volant small mammals in Southeast Asia from different families, with about 62 of those endemic to the region [5].

However, some species of small mammals are of public health concern as they are carriers of zoonotic diseases. For example, rodents are considered the main host of leptospires bacteria that cause leptospirosis [6,7,8,9,10]. Pathogenic *Leptospira* comprises eighteen identified species and more than 250 recognized pathogenic serovars, of which only the Icterohaemorrhagiae complex causes the most severe disease [11,12,13].

Leptospirosis outbreaks are common in the tropical region since this area provides favorable conditions such as high levels of rainfall and flooding, which are necessary for leptospires to survive [14]. Usually, exposure to pathogenic *Leptospira* depends on contact between humans and infected animals or contaminated environments or surfaces [15]. On the other hand, leptospirosis is common among individuals that come in frequent contact with contaminated surfaces, a process that could be easily activated during leisure activities in infected recreational areas [6].

To our knowledge, studies on the prevalence of pathogenic *Leptospira* in small mammals of Malaysia are limited to the urban areas [16,17], suburban areas [18,19], and oil-palm plantations [18], with few or none directly reporting the prevalence of pathogenic *Leptospira* in recreational areas, thereby limiting our understanding of the disease transmission potential in recreational areas. Only one study has reported pathogenic *Leptospira* spp. in water samples obtained from selected recreational areas of Terengganu [20]; however, there is a lack of information on the species composition of non-volant small mammals and the prevalence of pathogenic *Leptospira* spp. in Terengganu. Therefore, this study examines the species composition of non-volant small mammals and their pathogenic *Leptospira* prevalence in a recreational forest located in Kuala Berang, Hulu Terengganu, Malaysia, in an attempt to guide potential future interventions.

## 2. Methods

### 2.1. Study Area

The study was carried out in Hutan Lipur Sekayu (N04°57.85′, E102°56.71′), a recreational forest located in Kuala Berang, Hulu Terengganu, Malaysia (Figure 1), with an estimated population of 89,000 inhabitants of a low to moderate standard of living [21]. The recreational area has a total landmass of about 30 hectares and boasts facilities such as a waterfall, rest areas, public toilets, camping sites, playgrounds, and food stalls for the visitors.

### 2.2. Animal Trapping and Sampling Collection

We trapped animals using wire-mesh live traps (25 cm × 15 cm × 12 cm). The traps were randomly activated along the forest trails/streams with an approximate 10 m distance between traps. There were 20 trapping points with 5 live traps set at each trapping point for three consecutive nights, totaling 300 trapping efforts per trapping session. Overall, we conducted 10 trapping sessions September 2019 and October 2020. Ripe-banana-baited traps were set at dusk and checked at dawn each day, with baits replaced each day to maintain freshness. The captured animals were kept in an individual cloth bag and transported to the field station for further analysis. We used South-East Asia’s Mammals Field Guide to identify the captured animals [22].

All captured animals were humanely euthanized using diethyl ether and processed for kidney removal. Kidney from individual animal was placed in a sterile specimen container and stored in a Panasonic Ultra-Low Freezer at −86 °C until further investigation. All animal procedures were carried out according to the protocol previously described by Mills et al. [23].

### 2.3. Ethical Statement

This study was conducted in accordance with the Malaysian laws regarding ethics in research. The Ethics Review Committee Board of the Universiti Malaysia Terengganu gave the approval and permission to conduct research on small mammals with Project Number: UMT/JKEPHT/2019/30.

### 2.4. DNA Extraction

DNA extraction procedure was performed on kidney samples according to the technical manual for genomic DNA Purification/Extraction Kit (QIAamp DNA DNeasy Blood and Tissue Kit, Qiagen, Redwood, CA, USA). Briefly, kidney samples were rinsed with a sterile phosphate-buffered saline solution to reduce possible bacterial contaminants. A small piece of kidney tissue was cut (25 mg) and placed into a 1.5 mL micro-centrifuge tube containing 180 µL Buffer ATL and 20 µL proteinase K to break the cell membrane and release DNA into the solution. Then, the tube was incubated at 56 °C until samples were completely lysed.

A 200 µL Buffer AL was added and the samples incubated at 56 °C for 10 min. A volume of 200 µL ethanol (96–100%) was added to enhance the binding of DNA to silica. The precipitated DNA was filtered through a DNeasy mini spin column by centrifugation for 1 min at 8000 rpm. The flow-through was discarded, and the filtered DNA placed in a new spin column with 500 µL AW1 and AW2 buffer. The flow-through and collection tube were removed after 3 min of centrifuge at 14,000 rpm. DNA was eluted in a new 2 mL micro-centrifuge tube with 100 µL Buffer AE. The quality and the quantity of the DNA were recorded using Nanodrop™.

### 2.5. PCR Detection of Pathogenic Leptospira spp. and Sequencing

The DNA of *Leptospira* was detected by the amplification of 16S rRNA and *lipL32* gene [24,25,26]. The DNA used in PCR analysis was standardized to 20 ng/µL, using the manufacturer’s guidelines as contained in the Qiagen kit. We used primers 16S rRNA with forward sequence 5′-GGC GCG TCT TAA ACA TG-3′ and 16S rRNA with reverse 5′-GTG CCA GCA GCC GCG GTA A-3′. The LipL32 primer set used for detecting the presence of *lipL32* gene comprised the following: forward sequence of LipL32 -45F 5′-AAG CAT TAC CGC TTG TGG TG-3′ and the reverse sequence of LipL32 -286R 5′-GAA CTC CCA TTT CAG CGA TT-3′ [27]. The positive and negative controls were included in each PCR run. DNA of pathogenic *Leptospira*, *L. interrogans* serovar Copenhageni strain M20 was used as the positive control, while distilled water was used as the negative control. Amplification of the DNA was conducted in a total volume of 25 µL consisting of 12.5 µL ready-to-use Mastermix, 1 µL forward, 1 µL reverse primer, and 0.5 µL probes at a concentration of 10 µM, and it was diluted to a final volume of distilled water and 5 µL DNA templates. These protocols of amplification were enhanced according to the Eppendorf real-time PCR system and consisted of 2-min denaturation at 94 °C followed by 35 cycles of amplification at 94 °C for 3 s, 58 °C for 15 s, 72 °C for 1 min, and a final extension at 72 °C for 10 min. Afterwards, the reaction was stopped at 4 °C, and the results of the data were analyzed using the software provided by a real-time PCR system (Eppendorf realplex 2.2). Gel electrophoresis in a 1.0% TAE agarose gel stained was run to analyze the amplification of products, with results deemed only valid if the positive and negative controls produced the expected results. Samples were interpreted as positive for pathogenic *Leptospira* spp. if a band corresponding to 242 base-pair (bp) was obtained or otherwise interpreted as negative. The positive samples from PCR were sequenced using the ABI PRISM 377 sequencer (Applied Biosystems, Waltham, MA, USA) producing 99% sequence accuracy. Using bioinformatics tools, the sequences were aligned and deposited in the National Center for Biotechnology Information (NCBI), Bethesda, USA.

### 2.6. Data Analysis

We used descriptive analysis to report species diversity and abundance, and we analyzed the association between *Leptospira* positive individuals and species, age, and sex using Fisher exact test for small samples. All analyses were performed in R 4.0.0 version [28].

## 3. Results

We performed 3,000 trappings over ten trapping sessions. Out of this, 121 traps were either lost or damaged, thus leaving us with a total of 2,879 effective trapping efforts. Overall, we captured 45 individuals (1.56% trapping success) from 8 species, comprising of 30 (66.7%) rats, 10 (22.2%) tree shrews, and 5 (11.1%) squirrels (Table 1). The captured non-volant small mammals belonged to two orders (Scandentia and Rodentia) and three families (Tupaiidae, Muridae, and Sciuridae). The majority (67%) of the captured individuals were from the family Muridae, followed by Tupaiidae (22%) and then Sciuridae (11%).

Out of the 45 captured individuals, 9 (20%) from four species of rodents, and one species of tree shrew, tested positive for pathogenic *Leptospira* (Table 1). *R. tiomanicus* (n = 22) was the most dominant captured species, followed by *T. glis* (n = 10). Additionally, we observed a non-significant association (*p* = 0.89) in the gender of the nine positive individuals (Appendix A). In the same vein, the Fisher exact test showed no significant difference (*p* = 0.50) between the eight adults and one juvenile that tested positive for pathogenic *Leptospira*.

The DNA isolated from the kidneys of the 45 individuals had all undergone PCR. Gene *lipL32* was amplified in 9 out of the 45 individuals, as shown in (Figure 2). All nine isolates amplified by LipL32 primers were confirmed as pathogenic *Leptospira* spp.

The genetic identification by 16S rRNA sequencing exhibited an identity of more than 95% (Table 2). This table only listed six out of nine samples from PCR as the other three were considered false-positive. Our sequencing data were compared to the NCBI database, and we found the nearest serovar to be *Leptospira interrogans* serovar Canicola and others to be uncultured *Leptospira* sp.

## 4. Discussion

The 1.56% trapping success recorded in this study could be due to the numbers of small mammals living in the study area, or disturbances from tourists. Nevertheless, our finding of family Muridae being the most abundant in the study area is not surprising, considering they are the most abundant family of rodents, with more than 560 species representing 126 genera [29]. Their abundant status in the study area might also be because the majority of this family are forest and habitat generalists that will typically exploit any habitat type and thrive in a wide range of conditions [30]. Moreover, most of the habitat generalist species from this family feed on large groups of leaves, seeds, fruits, and roots that are usually available in the study area, thus enhancing their survival rate in the study area [31]. Apart from the family Muridae, family Tupaiidae (treeshrew) and Sciuridae (squirrel) were also recorded in this study, suggesting high diversity of rodent species in the study area. Both of these families share the same characteristics: they are diurnal in nature, with their main diet including insects and a wide variety of fruits and seeds [22], while they can also inhabit a wide range of forest types such as plantations, cultivated areas, and gardens.

The most dominant species recorded in this study is in line with the previous study by Rahim et al. [32], probably due to this species being habitat generalists, nocturnal species capable of inhabiting forested/agricultural areas that are capable of spending time in trees and similarly exploring the ground [22]. Moreover, they hide under fallen trees/branches in low woody vegetation. These attributes altogether probably contribute to their high capture rate.

The prevalence of pathogenic *Leptospira* in the five species is similar to the result of a previous study conducted in a recreational area by Yusof et al. [33], who found *Leptospira* in all species reported in the study except for *M. rajah*. This shows that the non-volant small mammals of Hutan Lipur Sekayu, especially shrews and rats, have the potential to maintain the pathogenic *Leptospira* transmission cycle in the study area. Additionally, the 20% (n = 9/45) prevalence rate of pathogenic *Leptospira* reported here is comparatively similar to that reported in a previous study (19.4%, n = 18/93) also from non-volant small mammals in recreational areas of Selangor, Malaysia [33]. Even though we could not statistically draw an inference from our results owing to the low sample size, our results show that *Leptospira* could be present in shrews and rats inhabiting the study area.

The sequencing data show that *R. tiomanicus*, *M. whiteheadi,* and *S. muelleri* could be infected with pathogenic *Leptospira* spp., of which *Leptospira interrogans* serovar Canicola is the most likely to cause leptospirosis. In Malaysia, this serovar has been isolated from water and soil samples from some campsites in Terengganu and Kelantan [34]. The serovar Canicola has also been reported to be the most vital pathogenic type of leptospires [35]. It is widely found in the urban areas infecting mostly canine [36], swine, and humans [35]. Our sequencing data show the presence of this serovar in rodent species occupying the recreational forest of Kuala Berang, Hulu Terengganu, Malaysia, and they are in accordance with previous studies in Malaysia, which also isolated *Leptospira interrogans* serovar Canicola among rodents, dogs, and cattle [19,37,38].

According to the Ministry of Health Malaysia [39], recreational areas are categorised as one of the most important hotspots for leptospirosis outbreaks after settlement areas. The somewhat poor garbage disposal by visitors at the centre could indirectly attract rats, which in-turn shed the pathogenic bacteria into the water/soil that could later infect visitors visiting the park [40].

In conclusion, for the first time, we have been able to provide useful information about the distribution of non-volant small mammals in a recreational forest of Kuala Berang, Hulu Terengganu, Malaysia, as well as their zoonotic potential or public health significance. Our results illustrate that non-volant small mammals, particularly shrews and rats, could maintain and transmit pathogenic *Leptospira* in the study area. Given none of the captured squirrels tested positive for *Leptospira* and considering little is known about their role as reservoirs of leptospirosis, their capability to transmit leptospirosis should be thoroughly investigated in future research. To reduce the risk of rodent-borne diseases and rodent proliferation, we recommend that effective public awareness programmes be encouraged in all recreational forests to increase tourists’ knowledge of leptospirosis and other zoonotic diseases. Additionally, we also suggest that appropriate waste management be implemented in the study area to reduce rodents’ access to food sources as this will reduce possible rat propagation and subsequent probable rat–human interactions in recreational areas.

## Figures and Tables

**Figure 1 pathogens-11-01300-f001:**
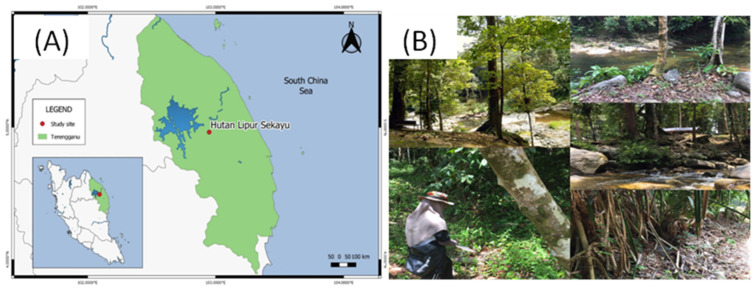
**Map of Hutan Lipur Sekayu, Terengganu, Malaysia.** (**A**) Location of the study site in Peninsular Malaysia, Terengganu state; (**B**) location of the trapping sites.

**Figure 2 pathogens-11-01300-f002:**
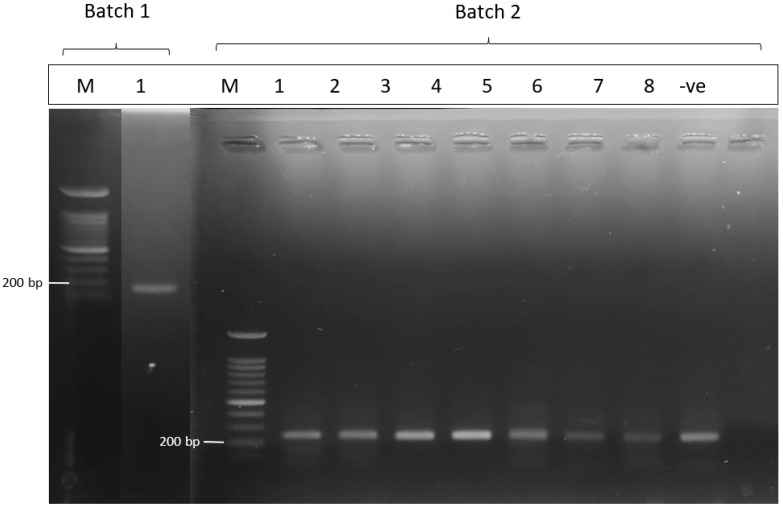
Gel electrophoresis image for positive pathogenic *Leptospira*. PCR was conducted into 2 batches. One out of 20 samples was amplified using LipL32 primers in the first batch. A total of eight out of 25 samples were positive in the second batch. The image shows the position of positive samples located between 200 bp and 300 bp. Lane M for both batches: DNA size marker, 100 bp DNA ladder; Lane 1: sample of kidney in batch 1; Lane 1–8: samples of kidneys in batch 2; and Lane ve: negative control.

**Table 1 pathogens-11-01300-t001:** Number of captured non-volant small mammal species and positive individuals in Hutan Lipur Sekayu recreational area.

Order and Family	Scientific Name	Common Name	Total Individuals Captured	Total Positive Individuals (%)
**Scandentia** **Tupaiidae**	*Tupaia glis*	Common Treeshrew	10	2 (20)
**Rodentia** **Muridae**	*Rattus tiomanicus*	Malaysian Wood Rat	22	4 (18.2)
	*Rattus rattus*	Black Rat	1	0 (0)
	*Maxomys rajah*	Rajah Spiny Rat	4	1 (25)
	*Maxomys whiteheadi*	Whitehead’s Spiny Rat	1	1 (100)
	*Sundamys muelleri*	Muller’s Giant Sunda Rat	2	1 (50)
**Rodentia** **Sciuridae**	*Collasciurus notatus*	Plaintain Squirrel	4	0 (0)
	*Sundasciurus tenuis*	Slender Squirrel	1	0 (0)
	Total		**45**	**9 (20)**

**Table 2 pathogens-11-01300-t002:** 16S rRNA sequencing from positive samples of pathogenic *Leptospira*.

Lane	Species	Percent ID	Query Cover	Accession No.
1 (Batch 1) (*R. tiomanicus*)	*Leptospira interrogans*	98.9%	99%	CP044513.1
1 (Batch 2) *M. whiteheadi*	*Leptospira interrogans*	97.6%	99%	KY356922.1
2 (*R. tiomanicus*)	*Leptospira interrogans*	98%	99%	KY356922.1
3 (*S. muelleri*)	Uncultured *Leptospira* sp.	97.5	98%	MG831575.1
5 (*R. tiomanicus*)	Uncultured *Leptospira* sp.	98.3%	97%	MG831575.1
6 (*R. tiomanicus*)	*Leptospira interrogans*	95.9%	99%	KY356922.1

## Data Availability

All data and the code used in this study are available in Zenodo under the Creative Common 4.0 license, accessible through https://doi.org/10.5281/zenodo.5156867 (accessed on 15 May 2021).

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
