# Peer review of "Prevalence of Pathogenic Leptospira spp. in Non-Volant Small Mammals of Hutan Lipur Sekayu, Terengganu, Malaysia"

_pathogens, 2022, doi:10.3390/pathogens11111300_

Round 1

Reviewer 1 Report

General remarks:

Given the complex epidemiology of leptospirosis, this topic is of enduring interest and contributes to our overall knowledge of animal reservoirs involved in the epidemiology of leptospirosis in Malaysia and in general. Although I do believe that research of small mammals and determination of their role in transmission of leptospirosis are very valuable and needed, I do have several suggestions, in terms of interpretation of the results that I would like to share with authors;

The low trapping success noted in this study may be due to the low numbers of small mammals living in the area (low abundance). It is clear that there is a high diversity of species, but that does not necessarily mean that there is a high abundance - it would be very interesting to give any data on the abundance of small mammals in the region, if they are available. Also, it is not clear from the data given whether there were differences in trapping success at different times of the year or in different locations? Perhaps some other rodents would be more attracted to a different type of bait (although banana is very commonly used). I suggest that the authors address in a sentence or two the possible reasons for this low trapping rate.

Although the number of animals captured was low, the overall prevalence was quite high (one in five animals was infected). You have found Leptospira only in shrews and rats, but not in squirrels. Perhaps that fact should also be mentioned in the text. There is evidence of the infection of certain species of squirrels, but very little is actually known about their role as reservoirs of leptospirosis.

Just a suggestion for the future is that you try to homogenize the kidney (it is not so difficult when frozen) and then take 25mg for DNA isolation.

Specific remarks:

Lines 27-29: Needs clarification. What exactly does "highest positive individual (44%, n= 4/9)” mean?

Lines 29-32: Please adjust conclusion to that that you have stated at the end of the manuscript. I would avoid the statement “are capable of maintaining and transmitting” and that part that “they are making this recreational area potential infestation ground for leptospirosis”. You have proven high diversity of non-volant small mammal species with high Leptospira carriage rates.

Lines 44-45: My suggestion is “However, some species of small mammals are of public health concern, as they are carriers of zoonotic diseases. For example, rodents are considered the main host of….”

Lines 46-47: you can omit following sentence from the text “Pathogenic Leptospira spp. is the causative agent of leptospirosis [10].

Line 48: Icterohaemorrhagiae is not a Leptospira species; it is a serogroup/serovar

Lines 204-205; see my comment for Lines 29-32

In new version of the text;

Table 1 and Table 2. are not displaying properly and can not see them!

Line; 164-165: What does it mean „cultured kidneys“? Have you been culturing kidneys or just isolating DNA and performing PCR?

Lines 220-229: not really sure that you can detect serovar with the methods that you have used.

Author Response

Responses to Reviewer’s Comments

We thank the Editor and Reviewer for their thoughtful comments and critiques which were useful in improving the manuscript and its presentation.

  1. The low trapping success noted in this study may be due to the low numbers of small mammals living in the area (low abundance). It is clear that there is a high diversity of species, but that does not necessarily mean that there is a high abundance - it would be very interesting to give any data on the abundance of small mammals in the region if they are available. Also, it is not clear from the data given whether there were differences in trapping success at different times of the year or in different locations? Perhaps some other rodents would be more attracted to a different type of bait (although banana is very commonly used). I suggest that the authors address in a sentence or two the possible reasons for this low trapping rate.

Response: The authors are grateful to the reviewer for these crucial points. We have slightly incorporated these comments in our manuscript. However, we used bait that is believed to be generally acceptable and palatable by rodents in the study area, and also we recorded similar capture success rate across the experiment.  

  1. Although the number of animals captured was low, the overall prevalence was quite high (one in five animals was infected). You have found Leptospiraonly in shrews and rats, but not in squirrels. Perhaps that fact should also be mentioned in the text. There is evidence of the infection of certain species of squirrels, but very little is actually known about their role as reservoirs of leptospirosis.

Response: Again, we thank the reviewer for this comment. We have mentioned this crucial point in the manuscript as suggested.  

  1. Just a suggestion for the future is that you try to homogenize the kidney (it is not so difficult when frozen) and then take 25mg for DNA isolation.

Response: This suggestion is well received, and it will be tried in our future research efforts

  1. Lines 27-29: Needs clarification. What exactly does "highest positive individual (44%, n= 4/9)” mean?

Response: We meant that Rattus tiomanicus had the highest infect individual, as compared with the other captured species

  1. Lines 29-32: Please adjust conclusion to that that you have stated at the end of the manuscript. I would avoid the statement “are capable of maintaining and transmitting” and that part that “they are making this recreational area potential infestation ground for leptospirosis”. You have proven high diversity of non-volant small mammal species with high Leptospiracarriage rates.

Response: The authors have modified the statement as advised. Thank you

  1. Lines 44-45: My suggestion is “However, some species of small mammals are of public health concern, as they are carriers of zoonotic diseases. For example, rodents are considered the main host of….”

Response: We have changed the statement as advised

  1. Lines 46-47: you can omit following sentence from the text “Pathogenic Leptospira is the causative agent of leptospirosis [10].

Response: We totally agree with the reviewer that this statement is redundant and we have taken it down. Thank you for pointing this out.

  1. Line 48: Icterohaemorrhagiae is not a Leptospiraspecies; it is a serogroup/serovar

Response: We have slightly modified the sentence to reflect the available numbers of recognized pathogenic serovars

  1. Lines 204-205; see my comment for Lines 29-32

Response: We have now modified the statement as suggested  

  1. In new version of the text;

Table 1 and Table 2. are not displaying properly and can not see them!

Response: We apologize for this technical error. The tables have now been reincorporated into the manuscript with an additional document containing only the tables.

  1. Line; 164-165: What does it mean „cultured kidneys“? Have you been culturing kidneys or just isolating DNA and performing PCR?

Response: We apologize for the use of a confusing term. We have changed the term ‘cultured kidney’ to ‘kidney’.

  1. Lines 220-229: not really sure that you can detect serovar with the methods that you have used.

Response: We have slightly modified the sentence to reflect the sequencing method that has been used to detect the serovar in this study.

Reviewer 2 Report

Dear authors,

The original manuscript entitled “Prevalence of pathogenic Leptospira spp. in non-volant small mammals of Hutan Lipur Sekayu, Terengganu, Malaysia” is suitably structured, developed and written by Shafie et al., appropriate English with a clear structure. They characterized the prevalence and species composition of pathogenic Leptospira spp. in non-volant small mammals of Hutan Lipur Sekayu, Terengganu. The results were interesting. They isolated 45 individuals from 8 species, with 9 individuals testing positive for pathogenic Leptospira. They found that Rattus tiomanicus was the most dominant isolated species. This is a very interesting and comprehensive scientific research on characterization of Leptospira isolates; however, there is some major points of view that should be considered. After developing the research, this manuscript should be re-considered for a critical evaluation.

-        Accession numbers of 16s rDNA sequences should be added into the supplementary data.

-        All gel images are not clear and visible. Please provide high-resolution images of gels. Also, all markers and PCR product bands should be determined in the image.

-        Discussion section is too limited. Please develop this section via comparison to other researches.

-        Significant level should be determined in Table 1.

-        Did you use the reference strain in this study? If yes please provide the reference number.

-        Setting up of the molecular characterization methodology is not clear. Please provide a better structure containing all references and gel images of the optimized methods in this study.

Author Response

31st October, 2022

Editorial Office,

Pathogens,

Dear Editorial Board Members:

Manuscript Number: Pathogens-1987253

Title:“Prevalence of pathogenic Leptospira spp. in non-volant small mammals of Hutan Lipur Sekayu, Terengganu, Malaysia

Responses to Reviewers’ Comments

We thank the Editor and Reviewer for their thoughtful comments and critiques which were useful in improving the manuscript and its presentation.

The original manuscript entitled “Prevalence of pathogenic Leptospira spp. in non-volant small mammals of Hutan Lipur Sekayu, Terengganu, Malaysia” is suitably structured, developed and written by Shafie et al., appropriate English with a clear structure. They characterized the prevalence and species composition of pathogenic Leptospira spp. in non-volant small mammals of Hutan Lipur Sekayu, Terengganu. The results were interesting. They isolated 45 individuals from 8 species, with 9 individuals testing positive for pathogenic Leptospira. They found that Rattus tiomanicus was the most dominant isolated species. This is a very interesting and comprehensive scientific research on characterization of Leptospira isolates; however, there is some major points of view that should be considered. After developing the research, this manuscript should be re-considered for a critical evaluation.

  1. Accession numbers of 16S rDNA sequences should be added into the supplementary data.

Response: Thank you for your comment. Accession numbers have been provided as listed in Table 2.

  1. All gel images are not clear and visible. Please provide high-resolution images of gels. Also, all markers and PCR product bands should be determined in the image.

Response: Thank you for your comment. We have included the high-resolution gel image, all markers and PCR product bands that show positive results are now shown in Figure 2. We have also labelled the gels as batch 1 and batch 2 to differentiate the time of the PCR analysis.

  1. Discussion section is too limited. Please develop this section via comparison to other researches.

Response: The authors have now improved the discussion section of the manuscript while also making comparison with previous studies

  1. Significant level should be determined in Table 1

Response: We have now included the p values in the manuscript. The authors are of the opinion that presenting this in the table will not be too appropriate, especially considering the type of statistical test used in this manuscript which only provides the general p-value rather than line-line p-value

  1. Did you use the reference strain in this study? If yes, please provide the reference number.

Response: Yes, the reference strain for positive control has been included.

  1. Setting up of the molecular characterization methodology is not clear. Please provide a better structure containing all references and gel images of the optimized methods in this study.

Response: In the methodology part, we have stated the references and methods for PCR (16S rRNA and lipL32 gene) and sequencing.

Reviewer 3 Report

introduction needs to be improved and the aim to be clear

Methods, At the line 137-143  "16S rRNA sequencing was performed for positive samples to identify Leptospira species [25]. Primers Lep1 and Lep2 which related  to 16S rRNA gene has forward sequence of 5’-GGA ACT GAG ACA CGG TCC AT -3’ 140 and reverse sequence of 5’- GCC TCA GCG TCA GTT TTA GG -3’. PCR was performed 141 using primers to amplify 412 base-pair (bp) fragment correspond to 16S rRNA genes"

How Lep1 and Lep2 which is related  to the 16S rRNA gene (it is an outer membrane  protein),

please rewrite the sequencing methods according to the primer used and the direction, and the sequencer type

please rewrite the conclusion

The manuscript needs extensive linguistic editing

Author Response

31st October, 2022

Editorial Office,

Pathogens,

Manuscript Number: Pathogens-1987253

Title:“Prevalence of pathogenic Leptospira spp. in non-volant small mammals of Hutan Lipur Sekayu, Terengganu, Malaysia

Responses to Reviewers’ Comments

We thank the Editor and Reviewer for their thoughtful comments and critiques which were useful in improving the manuscript and its presentation.

  1. Introduction needs to be improved and the aim to be clear

Response: Thank you for these useful comments. We have slightly improved the introduction as well as its presentation

  1. Methods, At the line 137-143 "16S rRNA sequencing was performed for positive samples to identify Leptospira species [25]. Primers Lep1 and Lep2 which related  to 16S rRNA gene has forward sequence of 5’-GGA ACT GAG ACA CGG TCC AT -3’ 140 and reverse sequence of 5’- GCC TCA GCG TCA GTT TTA GG -3’. PCR was performed 141 using primers to amplify 412 base-pair (bp) fragment correspond to 16S rRNA genes"

How Lep1 and Lep2 which is related to the 16S rRNA gene (it is an outer membrane protein)

Response: Thank you for your comment. We have removed these sentences to make it clearer, instead we have rewritten the methods for PCR and sequencing.

  1. Please rewrite the sequencing methods according to the primer used and the direction, and the sequencer type.

Response: We have added the sequencing method together with the sequencer used and forward-reverse primers for both 16S rRNA and lipL32 gene.

  1. Please rewrite the conclusion

Response: The authors have revised the conclusion as suggested

  1. The manuscript needs extensive linguistic editing

Response: We have now revised the use of English language of the manuscript as advised, and we are sincerely thankful for this suggestion which will no doubt aid the presentation and easy comprehension of our manuscript.

Round 2

Reviewer 2 Report

Dear authors,

All revisions have been addressed and this manuscript can be accepted for publication in its present form. 

Author Response

Responses to Reviewer’s Comments

The authors are grateful to the Editorial Team and the Reviewer for their thoughtful comments which were useful in improving our manuscript and its overall presentation.

Reviewer 3 Report

please provide another copy since the result has not appeared.

other comments are addressed correctly in the manuscript

Author Response

Responses to Reviewer’s Comments

The authors are grateful to the Editorial Team and the Reviewer for their thoughtful comments which were useful in improving our manuscript and its overall presentation.

  1. please provide another copy since the result has not appeared.

Response: We apologize for this technical error. The authors have now provided another copy of the manuscript (both word and pdf with the results/tables). Likewise, we have also uploaded the tables in a separate document.